# Guiding Program Synthesis by Learning to Generate Examples

**Larissa Laich, Pavol Bielik, Martin Vechev**
Department of Computer Science
ETH Zurich, Switzerland
llaich@ethz.ch, {pavol.bielik, martin.vechev}@inf.ethz.ch

## Abstract

A key challenge of existing program synthesizers is ensuring that the synthesized program generalizes well. This can be difficult to achieve as the specification provided by the end user is often limited, containing as few as one or two input-output examples. In this paper we address this challenge via an iterative approach that finds ambiguities in the provided specification and learns to resolve these by generating additional input-output examples. The main insight is to reduce the problem of selecting which program generalizes well to the simpler task of deciding which output is correct. As a result, to train our probabilistic models, we can take advantage of the large amounts of data in the form of program outputs, which are often much easier to obtain than the corresponding ground-truth programs.

## 1 Introduction

Over the years, program synthesis has been applied to a wide variety of different tasks including string, number or date transformations (Gulwani, 2011; Singh & Gulwani, 2012; 2016; Ellis et al., 2019; Menon et al., 2013; Ellis & Gulwani, 2017), layout and graphic program generation (Bielik et al., 2018; Hempel & Chugh, 2016; Ellis et al., 2019; 2018), data extraction (Barowy et al., 2014; Le & Gulwani, 2014; Iyer et al., 2019), superoptimization (Phothilimthana et al., 2016; Schkufza et al., 2016), code repair (Singh et al., 2013; Nguyen et al., 2013; D'Antoni et al., 2016), language modelling (Bielik et al., 2017), synthesis of data processing programs (Polosukhin & Skidanov, 2018; Nye et al., 2019) or semantic parsing Shin et al. (2019a). To capture user intent in an easy and intuitive way, many program synthesizers let its users provide a set of input-output examples $\mathcal{I}$ which the synthesized program should satisfy.

**Generalization challenge** A natural expectation of the end user in this setting is that the synthesized program works well even when $\mathcal{I}$ is severely limited (e.g., to one or two examples). Because of this small number of examples and the big search space of possible programs, there are often millions of programs consistent with $\mathcal{I}$. However, only a small number of them generalizes well to unseen examples which makes the synthesis problem difficult.

**Existing methods** Several approaches have provided ways to address the above challenge, including using an external model that *learns to rank* candidate programs returned by the synthesizer, modifying the search procedure by *learning to guide* the synthesizer such that it returns more likely programs directly, or *neural program induction* methods that replace the synthesizer with a neural network to generate outputs directly using a latent program representation. However, regardless of what other features these approaches use, such as conditioning on program traces (Shin et al., 2018; Ellis & Gulwani, 2017; Chen et al., 2019) or pre-training on the input data (Singh, 2016), they are limited by the fact that their models are conditioned on the initial, limited user specification.

**This work** We present a new approach for program synthesis from examples which addresses the above challenge. The key idea is to resolve ambiguity by iteratively strengthening the initial specification $\mathcal{I}$ with new examples. To achieve this, we start by using an existing synthesizer to find a candidate program $p_1$ that satisfies all examples in $\mathcal{I}$. Instead of returning $p_1$, we use it to find a distinguishing input $x^*$ that leads to ambiguities, i.e., other programs $p_i$ that satisfy $\mathcal{I}$ but produce different outputs $p_1(x^*) \neq p_i(x^*)$. To resolve this ambiguity, we first generate a set of candidate outputs for $x^*$, then use a neural model (which we train beforehand) that acts as an oracle and selects

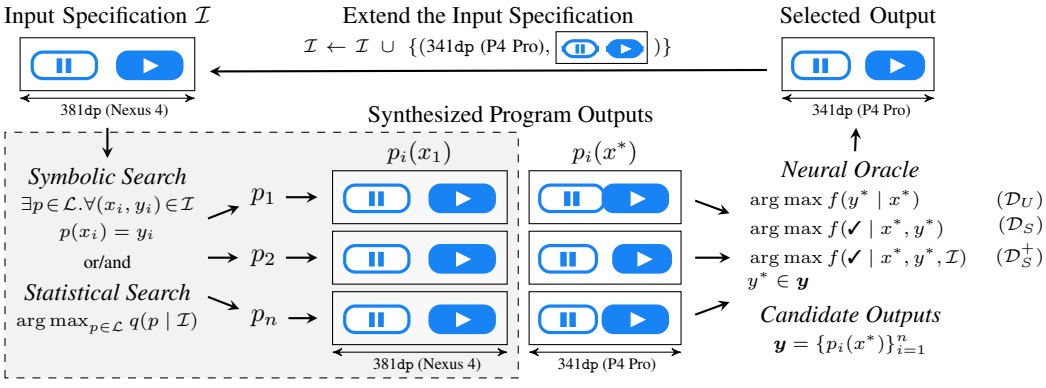

Figure 1: An overview of our approach which introduces a refinement loop around a black-box synthesizer that incrementally extends the input specification $\mathcal{I} = \{(x_1, y_1)\}$, which in this case contains a single example, with additional input-output examples until all ambiguities are resolved.

the most likely output, and finally, add $x^*$ and its predicted output to the input specification $\mathcal{I}$. The whole process is then repeated. These steps are similar to those used in Oracle Guided Inductive Synthesis (Jha et al., 2010) with two main differences: *(i)* we automate the entire process by learning the oracle from data instead of using a human oracle, and *(ii)* as we do not use a human oracle to produce a correct output, we need to ensure that the set of candidate outputs contains the correct one.

**Augmenting an existing Android layout synthesizer**    In this work we apply our approach to a state-of-the-art synthesizer, called `InferUI` (Bielik et al., 2018), that creates an Android layout program which represents the implementation of a user interface. Given an application design consisting of a set of views (e.g., buttons, images, text fields, etc.) and their location on the device screen, `InferUI` synthesizes a layout program that when rendered, places the views at that same location. Concretely, each input-output example $(x, y)$ consists of a device screen $x \in \mathbb{R}^4$ and a set of $n$ views $y \in \mathbb{R}^{n \times 4}$, all of which are represented using their coordinates in a two dimensional euclidean space. As an example, the input specification $\mathcal{I}$ shown in Figure 1 contains a single example with absolute view positions for a Nexus 4 device and the `InferUI` synthesizer easily finds multiple programs that satisfy it (dashed box). To apply our method and resolve the ambiguity, we find a distinguishing input $x^*$, in this case a narrower P4 Pro device, on which some of the candidate programs produce different outputs. Then, instead of asking the user to manually produce the correct output, we generate additional candidate outputs (to ensure that the correct one is included) and our learned neural oracle automatically selects one of these outputs (the one it believes is correct) and adds it to the input specification $\mathcal{I}$. In this case, the oracle selects output $p_2(x^*)$ as both buttons are correctly resized and the distance between them was reduced to match the smaller device width. In contrast, $p_1(x^*)$ contains overlapping buttons while in $p_n(x^*)$, only the left button was resized.

**Automatically obtaining real-world datasets**    An important advantage of our approach is that we reduce the problem of selecting which program generalizes well to the simpler task of deciding which output is correct. This is especially useful for domains, such as the Android layout synthesis, for which the output correctness depends mostly on properties of the output and not on the program used to generate it. As a result, obtaining a suitable training dataset can be easier as we do not require the hard to obtain ground-truth programs, for which currently no large real-world datasets exist (Shin et al., 2019b). In fact, it is possible to train the oracle using unsupervised learning only, with a dataset consisting of correct input-output examples $\mathcal{D}_U$. For example, in layout synthesis an autoencoder can be trained over a large number of views and their positions extracted from running real-world applications. However, instead of training such an unsupervised model, in our work we use $\mathcal{D}_U$ to automatically construct a supervised dataset $\mathcal{D}_S$ by labelling the samples in $\mathcal{D}_U$ as positive and generating a set of negative samples by adding suitable noise to the samples in $\mathcal{D}_U$. Finally, we also obtain the dataset $\mathcal{D}_{S+}$ that additionally includes the input specification $\mathcal{I}$. In the domain of Android layouts, although it is more difficult, such a dataset can also be collected automatically by running the same application on devices with different screen sizes.

**Our contributions** We present a new approach to address the ambiguity in the existing Android layout program synthesizer `InferUI` by iteratively extending the user provided specification with new input-output examples. The key component of our method is a learned neural oracle used to generate new examples trained with datasets that do not require human annotations or ground-truth programs. To improve generalization, `InferUI` already contains a probabilistic model that scores programs $q(p \mid \mathcal{I})$ as well as handcrafted robustness properties, achieving 35% generalization accuracy on a dataset of Google Play Store applications. In contrast, our method significantly improves the accuracy to 71% while using a dataset containing only correct and incorrect program outputs. We make our implementation and datasets available online at:

$$\text{https://github.com/eth-sri/guiding-synthesizers}$$

## 2  RELATED WORK

In this section we discuss the work most closely related to ours.

**Guiding program synthesis** To improve scalability and generalization of program synthesizers several techniques have been proposed that guide the synthesizer towards good programs. The most widely used approach is to implement a statistical search procedure which explores candidate programs based on some type of learned probabilistic model – log-linear models (Menon et al., 2013; Long & Rinard, 2016), hierarchical Bayesian prior (Liang et al., 2010), probabilistic higher order grammar (Lee et al., 2018) or neural network (Balog et al., 2017; Sun et al., 2018). Kalyan et al. (2018) also takes advantage of probabilistic models but instead of implementing a custom search procedure, they use the learned model to guide an existing symbolic search engine. In addition to approaches that search for a good program directly (conditioned on the input specification), a number of works guide the search by first selecting a high-level sketch of the program and then filling in the holes using symbolic (Ellis et al., 2018; Murali et al., 2017; Nye et al., 2019), enumerative or neural search (Bosnjak et al., 2017; Gaunt et al., 2016). A similar idea is also used by (Shin et al., 2018), but instead of generating a program sketch the authors first infer execution traces (or condition on partial traces obtained as the program is being generated (Chen et al., 2019)), which are then used to guide the synthesis of the actual program.

In comparison to prior work, a key aspect of our approach is to guide the synthesis by generating additional input-output examples that resolve the ambiguities in the input specification. Guiding the synthesizer in this way has several advantages – *(i)* it is interpretable and the user can inspect the generated examples, *(ii)* it can be used to extend any existing synthesizer by introducing a refinement loop around it, *(iii)* the learned model is independent of the actual synthesizer (and its domain specific language) and instead is focused only on learning the relation between likely and unlikely input-output examples, and *(iv)* often it is easier to obtain a dataset containing program outputs instead of a dataset consisting of the actual programs. Further, our approach is complementary to prior works as it treats the synthesizer as a black-box that can generate candidate programs. We also note that several prior works explore the design of sophisticated neural architectures that encode input-output examples (Sun et al., 2018; Devlin et al., 2017; Parisotto et al., 2017) and incorporating some of their ideas might lead to further improvements to our models presented in Section 4.

**Learning to rank** To choose among all programs that satisfy the input specification, existing program synthesizers select the syntactically shortest program (Liang et al., 2010; Polozov & Gulwani, 2015; Raychev et al., 2016), the semantically closest program to a reference program (D'Antoni et al., 2016) or a program based on a learned scoring function (Liang et al., 2010; Mandelin et al., 2005; Singh & Gulwani, 2015; Ellis & Gulwani, 2017; Singh, 2016). Although the scoring function usually extracts features only from the synthesized program, some approaches also take advantage of additional information – Ellis & Gulwani (2017) trains a log-linear model using a set of handcrafted features defined over program traces and program outputs while Singh (2016) leverages unlabelled data by learning common substring expressions shared across the input data.

Similar to prior work, our work explores various representations over which the model is learned. Because we applied our work to a domain where outputs can be represented as images (rather than strings or numbers), to achieve good performance we explore different types of models (i.e., convolutional neural networks). Further, we do not assume that the synthesizer can efficiently enumerate

all programs that satisfy the input specification as in Ellis & Gulwani (2017); Singh (2016). For such synthesizers, applying a ranking of the returned candidates will often fail since the correct program is simply not included in the set of synthesized programs. Therefore, the neural oracle is defined over program outputs instead of actual programs. This reduces the search space for the synthesizer as well as the complexity of the machine learning models.

**Neural program induction**   Devlin et al. (2017) and Parisotto et al. (2017), as well as related line of work on neural machines (Graves et al., 2016; Reed & de Freitas, 2016; Bošnjak et al., 2017; Chen et al., 2018), explore the design of end-to-end neural approaches that generate the program output for a new input without the need for an explicit search. In this case the goal of the neural network is not to find the correct program explicitly, but rather to generate the most likely output for a given input based on the input specification. These approaches can be integrated in our work as one technique for generating a set of candidate outputs for a given distinguishing input instead of obtaining them using a symbolic synthesizer. However, the model requirements in our work are much weaker – it is enough if the correct output is among the top $n$ most likely candidates rather than requiring 100% precision for all possible inputs as in program induction.

## 3   LEARNING TO GENERATE NEW INPUT-OUTPUT EXAMPLES

Let $\mathcal{I} = \{(x_i, y_i)\}_{i=1}^N$ denote the input specification consisting of user provided input-output examples. Further, assume we are given an existing synthesizer which can find a program $p$ satisfying all examples in $\mathcal{I}$, i.e., $\exists p \in \mathcal{L}, \forall (x_i, y_i) \in \mathcal{I}. \, p(x_i) = y_i$, where $p(x_i)$ is the output obtained by running program $p$ on input $x_i$ and $\mathcal{L}$ is a hypothesis space of valid programs. To reduce clutter, we use the notation $p \models \mathcal{I}$ to denote that $p$ satisfies all examples in $\mathcal{I}$. We extend the synthesizer such that a program $p$ not only satisfies all examples in $\mathcal{I}$ but also generalizes to unseen examples as follows:

1. Generate a candidate program $p_1 \models \mathcal{I}$ that satisfies the input specification $\mathcal{I}$.

2. Find a *distinguishing input* $x^*$, a set of programs $p_2, \ldots, p_n$ that satisfy the input specification $\mathcal{I}$ but produce different outputs when evaluated on $x^*$, and define candidate outputs as $\boldsymbol{y} = \{p_1(x^*), p_2(x^*), \cdots, p_n(x^*)\}$. If no distinguishing input $x^*$ exists, return program $p_1$.

3. Query an oracle to determine the correct output $y^* \in \boldsymbol{y}$ for the input $x^*$.

4. Extend the input specification with the *distinguishing input* and its corresponding output $\mathcal{I} \leftarrow \mathcal{I} \cup \{(x^*, y^*)\}$ and continue with the first step.

**Finding a distinguishing input**   To find the distinguishing input $x^*$ we take advantage of existing symbolic synthesizers by asking the synthesizer to solve $\exists x^* \in \mathcal{X}, p_2 \in \mathcal{L}. \, p_2 \models \mathcal{I} \wedge p_2(x^*) \neq p_1(x^*)$, where $\mathcal{X}$ denotes a set of valid inputs. The result is both a distinguishing input $x^*$ as well as a program $p_2$ that produces a different output than $p_1$. Programs $p_1$ and $p_2$ form the initial sequence of candidate outputs $\boldsymbol{y} = [p_1(x^*); p_2(x^*)]$ which is extended until the oracle is confident enough that $\boldsymbol{y}$ contains the correct output (described later in this section).

To make our approach applicable to any existing synthesizer, including those that can not solve the above satisfiability query directly (e.g., statistical synthesizers), we note that the following sampling approach can also be applied: first, use the synthesizer to generate the top $n$ most likely programs, then randomly sample a valid input $x^*$ not in the input specification, and finally check if that input leads to ambiguities by computing the output of all candidate programs.

**Finding candidate outputs**   To extend $\boldsymbol{y}$ with additional candidate outputs once the distinguishing input $x^*$ is found, three techniques can be applied: *(i)* querying the synthesizer for another program with a different output: $\exists p \in \mathcal{L}. \, p \models \mathcal{I} \wedge \forall_{y_i \in \boldsymbol{y}} p(x^*) \neq y_i$, *(ii)* sampling a program induction model $P(y \mid x^*, \mathcal{I})$ and using the synthesizer to check whether each sampled output is valid, or *(iii)* simply sampling more candidate programs, running them on $x^*$ and keeping the unique outputs. It is possible to use the second approach as we are only interested in the set of different outputs, rather than the actual programs. The advantage of *(i)* is that it is simple to implement for symbolic synthesizers and is guaranteed to find different outputs if they exist. In contrast, *(ii)* has the potential to be faster as it avoids calling the synthesizer and works for both statistical and symbolic synthesizers. Finally, *(iii)* is least effective, but it does not require pretraining and can be applied to any synthesizer.

**Neural oracle** The key component of our approach is a neural oracle which selects the correct program output from a set of candidate outputs $\boldsymbol{y}$. Formally, the neural oracle is defined as $\arg\max_{y^* \in \boldsymbol{y}} f_\theta(\checkmark \mid x^*, y^*, \mathcal{I})$, where $f$ is a function with learnable parameters $\theta$ (in our case a neural network) that returns the probability of the output $y^*$ being correct given the input $x^*$ and the input specification $\mathcal{I}$. We train the parameters $\theta$ using a supervised dataset $\mathcal{D}_{S+} = \{(\checkmark, x_i, y_i, \mathcal{I}_i)\}_{i=1}^{N} \cup \{(\boldsymbol{X}, x_j, y_j, \mathcal{I}_j)\}_{j=1}^{M}$ which, for a given distinguishing input $x^*$ and input specification $\mathcal{I}$, contains both the correct ($\checkmark$) as well as the incorrect ($\boldsymbol{X}$) outputs. Because it might difficult to obtain such a dataset in practice, we also define a simpler model $f_\theta(\checkmark \mid x^*, y^*)$ that is trained using a supervised dataset $\mathcal{D}_S = \{(\checkmark, x_i, y_i)\}_{i=1}^{N} \cup \{(\boldsymbol{X}, x_j, y_j)\}_{j=1}^{M}$ which does not include the input specification. In the extreme case, where the dataset contains only the correct input-output examples $\mathcal{D}_U = \{(x_i, y_i)\}_{i=1}^{N}$, we define the oracle as $f_\theta(y^* \mid x^*)$. Even though the dataset does not contain any labels, we can still train $f$ in an unsupervised manner. This can be achieved for example by splitting the output into smaller parts $y_i = y_i^1 \cdots y_i^t$ (such as splitting a word into characters) and training $f$ as an unsupervised language model. Alternatively, we could also train an autoencoder that first compresses the output into a lower dimensional representation, with the loss corresponding to how well it can be reconstructed. To achieve good performance, the architecture used to represent $f$ is tailored to the synthesis domain at hand, as discussed in the next section.

**Dynamically controlling the number of generated candidate outputs** Since generating a large number of candidate outputs is time consuming, we allow our models to dynamically control the number of sampled candidates for each distinguishing input $x^*$. That is, instead of generating all candidate outputs $\boldsymbol{y}$ first and only then querying the neural oracle to select the most likely one, we query the oracle after each generated candidate output and let the model decide whether more candidates should be generated. Concretely, we define a threshold hyperparameter $t \in \mathbb{R}^{[0,1]}$ which is used to return the first candidate output $y^*$ for which the probability of the output being correct is above this threshold. Then, only if there are no candidate outputs with probability higher that $t$, we return $\arg\max$ of all the candidates. Note for $t = 1$ this formulation is equivalent to returning the $\arg\max$ of all the candidates while for $t = 0$, it corresponds to always returning the first candidate, regardless of its probability of being correct. We show the effect of different threshold values as well as the number of generated candidate outputs in Section 5.

## 4 Instantiation of our approach to Android layout synthesis

In this section we describe how to apply our approach to the existing Android layout program synthesizer `InferUI` (Bielik et al., 2018). Here, the input $x \in \mathbb{R}^4$ defines the absolute positions of the top left and bottom right coordinates of a given device screen while the output $y \in \mathbb{R}^{n \times 4}$ consists of $n$ views and their absolute positions.

**Finding a distinguishing input and candidate outputs** Because `InferUI` uses symbolic search, finding a distinguishing input and candidate outputs is encoded as a logical query solved by the synthesizer as described in Section 3. However, instead of synthesizing the layout program containing correct outputs of all the views at once (as done in `InferUI`), we run the synthesizer $n$ times, each time predicting the correct output for only a single view (starting from the largest view) which is then added as an additional input-output example (we provide a concrete example in Appendix C.4). This is necessary since there are exponentially many combinations of the view positions when considering all the views at once and the `InferUI` synthesizer is not powerful enough to include the correct one in the set of candidate outputs (e.g., for samples with more than 10 views in less than 4%). The advantage of allowing the position of only a single view to change, while fixing the position of all prior views, is that the correct candidate output is much easier to include. The disadvantage is that the neural oracle only has access to partial information (consisting of the prior view positions) and therefore performs a sequence of greedy predictions rather than optimizing all view positions jointly.

**Neural oracle** Because the input $x$ has the same dimensions as each view, we encode it as an additional view in all our network architectures. In Figure 2 we show three different neural architectures that implement the oracle function $f$, each of which uses a different way to encode the input-output example into a hidden representation followed by a fully-connected ReLU layer and a softmax that computes the probability that the output is correct. In the following, we describe the architecture of all the models. The formal feature definitions are included in Appendix A.

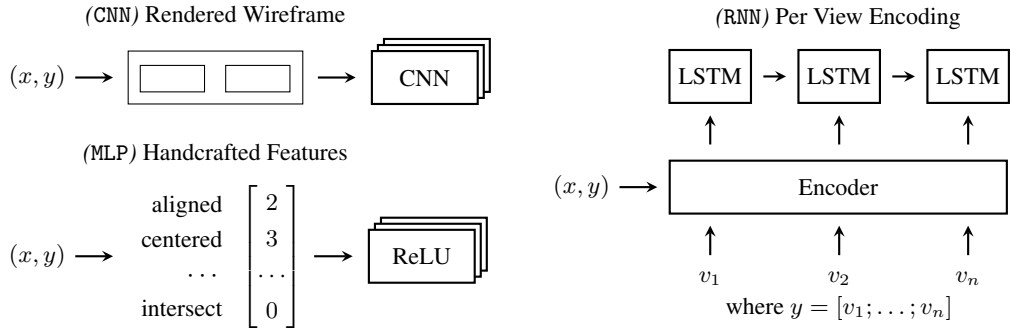

Figure 2: Illustration of three different models used to implement the neural oracle – CNN, RNN and MLP. All models include a fully-connected ReLU layer and a softmax layer (not shown in the image) which computes the probability that the output $y$ is correct. Note, that the features in the MLP model are shown unnormalized, that is, centered $= 3$ denotes that three views are centered.

In (CNN), the output is converted to an image (each view is drawn as a rectangle with a 1px black border on a white background) and used as an input to a convolutional neural network (CNN) with 3 convolutional layers with $5 \times 5$ filters of size 64, 32 and 16 and max pooling with kernel size 2 and stride 2. To support outputs computed for different screen sizes, the image dimensions are slightly larger than the largest device size. We regularize the network during training by positioning the outputs with a random offset such that they are still fully inside the image, instead of placing them in the center. This is possible as the image used as input to the CNN is larger than the device size. We provide visualization of this regularization in Appendix C.2.

In (MLP), the output is transformed to a normalized feature vector and then fed into a feedforward neural network with 3 hidden layers of size 512 with ReLU activations. To encode the properties of a candidate output $y$, we use high-level handcrafted features adapted from InferUI such as the number of view intersections or whether the views are aligned or centered. By instantiating the features for both horizontal and vertical orientation we obtain a vector of size 30, denoted as $\varphi_{\text{MLP}}(x^*, y^*)$, which is used as input to the neural network. For the model $f(\checkmark \mid x^*, y^*, \mathcal{I})$ the network input is a concatenation of output features (as before) $\varphi_{\text{MLP}}(x^*, y^*)$, features of each sample in the input specification $[\varphi_{\text{MLP}}(x, y)]_{(x,y) \in \mathcal{I}}$, and their difference $[\varphi_{\text{MLP}}(x^*, y^*) - \varphi_{\text{MLP}}(x, y)]_{(x,y) \in \mathcal{I}}$. This difference captures how the views have been resized or moved between the devices with different screen dimensions. This allows the model to distinguish between outputs that are all likely when considered in isolation but not when also compared to examples in $\mathcal{I}$ (as illustrated in Appendix C.3).

In (RNN), we use the fact that each output consists of a set of views $y^* = [v_1; \dots; v_n]$ by using an encoder to first compute a hidden representation of each view. These are then combined with a LSTM to compute the representation of the whole output. To encode a view $v_i$, we extract pairwise feature vectors with all other views (including the input) $\varphi_{\text{RNN}}(v_i, x^*, y^*) = [\phi(v_i, v_j)]_{v_j \in \{x^*\} \cup y^* \setminus v_i}$ and combine them using a LSTM. Here, $\phi : \mathbb{R}^4 \times \mathbb{R}^4 \to \mathbb{R}^n$ is a function that extracts $n$ real valued features from a pair of views. For each pair of views, we apply 11 simple transformations of the view coordinates, capturing their distance (4 vertical, 4 horizontal), the size difference (in width and height) and the ratio of the aspect ratios. For the model $f(\checkmark \mid x^*, y^*)$ we additionally use 17 high-level features computed for each view that are adapted from InferUI. When using $f(\checkmark \mid x^*, y^*, \mathcal{I})$, these additional high-level features are not required and instead we only use the 11 simple transformations combined in the same way as for the MLP model. That is, by concatenating $\varphi_{\text{RNN}}(v_i, x^*, y^*)$, $[\varphi_{\text{RNN}}(v_i, x, y)]_{(x,y) \in \mathcal{I}}$ and their difference $[\varphi_{\text{RNN}}(v_i, x^*, y^*) - \varphi_{\text{RNN}}(v_i, x, y)]_{(x,y) \in \mathcal{I}}$.

**Datasets** To train our models we obtained three datasets $\mathcal{D}_U$, $\mathcal{D}_S$ and $\mathcal{D}_S^+$, each containing an increasing amount of information at the expense of being harder to collect.

The unsupervised $\mathcal{D}_U = \{(x_i, y_i)\}_{i=1}^N$ is the simplest dataset and contains only positive input-output samples obtained by sampling $\approx 22,000$ unique screenshots (including the associated metadata of all the absolute view positions) of Google Play Store applications taken from the Rico dataset (Deka et al., 2017). Since the screenshots always consist of multiple layout programs combined together,

Table 1: Generalization accuracy of the existing `InferUI` synthesizer.

| `InferUI`
(Bielik et al., 2018) | $p \models \mathcal{I}$
*baseline* | $\arg\max_{p \models \mathcal{I}} q(p \mid \mathcal{I})$
*+ probabilistic model* | $\arg\max_{p \models \mathcal{I} \wedge \phi(p)} q(p \mid \mathcal{I})$
*+ probabilistic & robustness model* |
|---|---|---|---|
| Accuracy | 15.5% | 24.7% | 35.2% |

we approximate the individual programs by sorting the views in decreasing order of their size and taking a prefix of random length (of up to 30 views). For all of the datasets, we deduplicate the views that have the same coordinates and filter out views with a negative width or height.

The supervised $\mathcal{D}_S = \{(\checkmark, x_i, y_i)\}_{(x_i,y_i)\in\mathcal{D}_U} \cup \{\bigcup_{j=1}^{\leq 15}\{(\times, x_i, y_i + \epsilon_{ij})\}\}_{(x_i,y_i)\in\mathcal{D}_U}$ contains both correct and incorrect input-output examples. In our work this dataset is produced synthetically from $\mathcal{D}_U$ by extending it with incorrect samples. Concretely, the positive samples correspond to those in the dataset $\mathcal{D}_U$ and for each positive sample we generate up to 15 negative samples by applying a transformation $\epsilon_{ij}$ to the correct output. The transformations considered in our work are sampled from the common mistakes the synthesizer can make – resizing a view, shifting a view horizontally, shifting a view vertically or any combination of the above.

The supervised dataset is $\mathcal{D}_{S+} = \{(\checkmark, x_i, y_i, \mathcal{I}_i)\}_{i=1}^N \cup \{(\times, x_j, y_j, \mathcal{I}_j)\}_{j=1}^M$, where each $\mathcal{I}_i$ contains the same application rendered on multiple devices. We downloaded the same applications as used in the `Rico` dataset from the Google Play Store and executed them on three Android emulators with different device sizes. The number of valid samples is $\approx 600$ since not all applications could be downloaded, executed or produced the same set of views (or screen content) when executed on three different devices. The negative examples are generated by running the synthesizer with the input specification $\mathcal{I}$ containing a single sample and selecting up to 16 outputs that are inconsistent with the ground-truth output for the other devices.

## 5 EVALUATION

We evaluate our approach by applying it to an existing Android layout synthesizer called `InferUI` (Bielik et al., 2018) as described in Section 4. `InferUI` is a symbolic synthesizer which encodes the synthesis problem as a set of logical constraints that are solved using the state-of-the-art SMT solver Z3 (De Moura & Bjørner, 2008). To improve generalization, `InferUI` already implements two techniques – a probabilistic model that selects the most likely program among those that satisfy the input specification, and a set of handcrafted robustness constraints $\phi(p)$ that prevent synthesizing layouts which violate good design practices. We show that even if we disable these two optimizations and instead guide the synthesizer purely by extending the input specification with additional input-output examples, we can still achieve an accuracy increase from 35% to 71%.

In all our experiments, we evaluate our models and `InferUI` on a test subset of the $\mathcal{D}_{S+}$ dataset which contains 85 Google Play Store applications, each of which contains the ground truth of the absolute view positions on three different screen dimensions – 1400×2520, 1440×2560 and 1480×2600. We use one screen dimension as the input specification $\mathcal{I}$, the second as the distinguishing input and the third one only to compute the generalization accuracy. The generalization accuracy of a synthesized program $p \models \mathcal{I}$ is defined as the percentage of views which the program $p$ renders at the correct position.

**InferUI Baseline** To establish a baseline, we run `InferUI` in three modes as shown in Table 1. The *baseline* mode returns the first program that satisfies the input specification, denoted as $p \models \mathcal{I}$, and achieves only 15.5% generalization accuracy. In the second mode the synthesizer returns the most likely program according to a *probabilistic model* $q(p \mid \mathcal{I})$ which leads to an improved accuracy of 24.7%. The third mode additionally defines a set of robustness properties $\phi(p)$ that the synthesized program needs to satisfy, which together with the probabilistic model achieve 35.2% accuracy. The generalization accuracy of all `InferUI` models is relatively low as we are using a challenging dataset where each sample contains on average 12 views. Note however, that this is expected since increasing the number of views leads to an exponentially larger hypothesis space.

Table 2: Generalization accuracy of different models used as neural oracle in our approach.

| Training Dataset | Model | Accuracy | | | |
|:---:|:---:|:---:|:---:|:---:|:---:|
| | | MLP | CNN | RNN | RNN + CNN |
| $\mathcal{D}_S$ | $f(\checkmark \mid x^*, y^*)$ | 14.3% | 14.2% | 23.8% | **32.1%** |
| $\mathcal{D}_{S+}$ | $f(\checkmark \mid x^*, y^*, \mathcal{I})$ | 20.7% | 33.2% | 63.4% | **71.0%** |

Table 3: Effect of the threshold $t$ and the maximum number of generated candidate outputs $|\boldsymbol{y}|$ on the accuracy and the average number of generated candidate outputs per view (shown in brackets).

| | | $t = 0.9$ | | | $t = 1$ |
|:---:|:---:|:---:|:---:|:---:|:---:|
| Model | | $|\boldsymbol{y}| \leq 4$ | $|\boldsymbol{y}| \leq 9$ | $|\boldsymbol{y}| \leq 16$ | $|\boldsymbol{y}| = 16$ |
| $f(\checkmark \mid x^*, y^*, \mathcal{I})$ | RNN + CNN | 40.8% (2.8) | 68.3% (4.7) | **71.3% (6.0)** | 71.0% (14.7) |

Further, to establish an upper bound on how effective a candidate ranking approach can be, we query the synthesizer for up to 100 different candidate programs (each producing a unique output) and check how often the correct program is included. While for small samples (with up to 4 views) the correct program is almost always included, for samples with 6 views it is among the synthesized candidates in only 30% of the cases and for samples with more than 10 views in less than 4%. Sampling more outputs will help only slightly as increasing the number of views would require generating exponentially more candidates.

**Our Work**   We apply our approach to the `InferUI` synthesizer by iteratively generating additional input-output examples that strengthen the input specification. The specification initially contains absolute positions of all the views for one device and we extend the specification by adding one view at a time (rendered on a different device) as described in Section 4. In the experiments presented here we focus on evaluating the overall improvement of the `InferUI` synthesizer extended with our approach. We provide additional experiments that evaluate the effectiveness of the neural oracle as well as an ablation study in Appendix B.

*Generalization Accuracy*   The results of our approach instantiated with various neural oracle models are shown in Table 2. The best model trained on the dataset $\mathcal{D}_S$ has almost the same accuracy as `InferUI` with all its optimizations enabled. This means that it is possible to replace existing optimizations and handcrafted robustness constraints by training on an easy to obtain dataset consisting of correct outputs and their perturbations. More importantly, when training on the harder to obtain dataset $\mathcal{D}_{S+}$, the generalization accuracy more than doubles to 71% since the model can also condition on the input specification $\mathcal{I}$. However, the results also show that the design of the model using the neural oracle is important for achieving good results. In particular, both `MLP` models achieve poor accuracy since the high level handcrafted features adapted from the `InferUI` synthesizer are not expressive enough to distinguish between correct and incorrect outputs. The `CNN` models achieve better accuracy but are limited for the opposite reason, they try to learn all the relevant features from the raw pixels which is challenging as many features require pixel level accuracy across large distances (e.g., whether two views in the opposite parts of the screen are aligned or centered). The `RNN` model performs the best, especially when also having access to the input specification. Even though it also processes low level information, such as distance or size difference between the views, it uses a more structured representation that first computes individual view representations that are combined to capture the whole output.

*Number of Candidate Outputs*   We show the effect of different threshold values $t$ used by the neural oracle to dynamically control whether to search for more candidate outputs as well as the maximum number of candidate outputs in Table 3. We can see that using the threshold both slightly improves the accuracy ($+0.3\%$) but more importantly, significantly reduces the average number of generated candidate outputs from $14.7$ to $6.0$ using the same number of maximum generated outputs $|\boldsymbol{y}| = 16$.

*Incorporating User Feedback*   Even though our approach significantly improves over the `InferUI` synthesizer, it does not achieve perfect generalization accuracy. This is because for many synthesizers the perfect generalization is usually not achievable – the correct program and its outputs depends on a user preference, which is only expressed as severely underspecified set of input-output examples. For example, for a given input specification there are often multiple good layout programs that do not violate any design guidelines and which one is chosen depends on a particular user. To achieve 100% in practice, we perform an experiment where the user can inspect the input-output examples generated by our approach and correct them if needed. Then, we simply count how many corrections were required. The applications in our dataset have on average 12 views and for our best model, no user corrections are required in 30% of the cases and in 27%, 15%, 12%, 5% of the cases the user needs to provide 1, 2, 3 or 4 corrections, respectively. In contrast, `InferUI` with all optimizations enabled requires on average twice as many user corrections and achieves perfect generalization (i.e., zero user corrections) in only 3.5% of the cases.

## 6   CONCLUSION

In this work we present a new approach to improve the generalization accuracy of existing program synthesizers. The main components of our method are: *(i)* an existing program synthesizer, *(ii)* a refinement loop around that synthesizer, which uses a neural oracle to iteratively extend the input specification with new input-output examples, and *(iii)* a neural oracle trained using an easy to obtain dataset consisting of program outputs. To show the practical usefulness of our approach we apply it to an existing Android layout synthesizer called `InferUI` (Bielik et al., 2018) and improve its generalization accuracy by $2\times$, from 35% to 71%, when evaluated on a challenging dataset of real-world Google Play Store applications.

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

APPENDIX

We provide three appendices. Appendix A contains an in-depth description of the feature transformations for the MLP and RNN models. Appendix B contains an additional ablation experiment as well as results from the oracle evaluation. Appendix C provides visualizations of positive and negative candidate outputs, the CNN regularization as well as outputs from the end-to-end synthesis.

## A    FEATURE DEFINITIONS

In this section we describe in detail the feature transformations used in Section 4. As mentioned in Section 4 each view consists of the 4 coordinates: $x_l$ ($x_{left}$), $y_t$ ($y_{top}$), $x_r$ ($x_{right}$), $y_b$ ($y_{bottom}$). We use $v.w$ for the width, $v.h$ for the height and $v.r$ for the aspect ratio ($v.w/v.h$) of view $v$.

### A.1    MLP

In Table 4 we define the 7 feature types that lead to a vector of size 30 used in the MLP model. All the features are normalized (divided) by the factor in the normalization column.

Table 4: Feature definitions for the MLP.

| Size | Name | Description | Normalization |
|---|---|---|---|
| 1 | Number of off-screen views | - | $|V|$ |
| 3 | Number of views with a specific aspect ratio | Instantiated for the ratios 1, [3/4, 4/3] and [9/16, 9/16] | $|V|$ |
| 1 | Number of view intersections | Pairwise comparison if two views intersect (without counting fully contained views.) | $|V|^2$ |
| 1 | Number of views which have the same dimension | The same dimension is defined as: $v_1.w = v_2.w \wedge v_1.h = v_2.h$ | $|V|^2$ |
| 1 vertical + 1 horizontal | Number of view alignments | Pairwise comparison if two views align (e.g. for vertical alignment: $v_1.x_l = v_2.x_l \vee v_1.x_l = v_2.x_r \vee v_1.x_r = v_2.x_l \vee v_1.x_r = v_2.x_r$ ) | $|V|^2$ |
| 9 vertical + 9 horizontal | Number of views with a specific margin to another view | Instantiated for the margins 0, 16, 28, 32, 40, 48, 60, 64 and 96. | $|V|^2$ |
| 1 vertical + 1 horizontal | Number of centered views | Pairwise comparison if one view is centered within another view. | $|V|^2$ |
| 1 vertical + 1 horizontal | Number of centered views between two different views | Comparison if one view is centered between 2 other views. | $|V|^3$ |

### A.2    RNN

We formally define the 11 transformations used as the pairwise view feature vector $\phi\colon \mathbb{R}^4 \times \mathbb{R}^4 \to \mathbb{R}^n$ in the RNN model. These features capture properties like the view's distance to the other view or the size difference. The feature vectors extracted for each pair of views are combined to a fixed length representation by passing them through the LSTM, in the decreasing order of view size. In Figure 3 all the properties are listed and visualized.

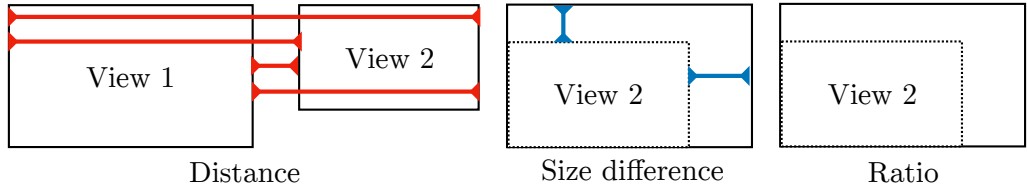

| Name | Description | Formula | | |
|------|-------------|---------|---|---|
| Distance | Horizontal distance from one view to another view | $d_{ll}(v_1, v_2)$ | = | $v_1.x_l - v_2.x_l$ |
| | | $d_{lr}(v_1, v_2)$ | = | $v_1.x_l - v_2.x_r$ |
| | | $d_{rl}(v_1, v_2)$ | = | $v_1.x_r - v_2.x_l$ |
| | | $d_{rr}(v_1, v_2)$ | = | $v_1.x_r - v_2.x_r$ |
| | Vertical distance from one view to another view | $d_{tt}(v_1, v_2)$ | = | $v_1.y_t - v_2.y_t$ |
| | | $d_{tb}(v_1, v_2)$ | = | $v_1.y_t - v_2.y_b$ |
| | | $d_{bt}(v_1, v_2)$ | = | $v_1.y_b - v_2.y_t$ |
| | | $d_{bb}(v_1, v_2)$ | = | $v_1.y_b - v_2.y_b$ |
| Size difference | Size difference of two views | $s_w(v_1, v_2)$ | = | $v_1.w - v_2.w$ |
| | | $s_h(v_1, v_2)$ | = | $v_1.h - v_2.h$ |
| Ratio | Relation of the aspect ratio | $r(v_1, v_2)$ | = | $v_1.r / v_2.r$ |

Figure 3: Definition of the pairwise view feature vector $\phi\colon \mathbb{R}^4 \times \mathbb{R}^4 \to \mathbb{R}^n$ used in the RNN model (bottom) and their visualization (top).

To guide the model for $f(\checkmark \mid x^*, y^*)$, we defined 17 more abstract features, adapted from `InferUI`, which are defined using the simple transformations shown in Figure 3. Concretely, we define the following 17 features:

- For alignments (8 features: 4 horizontal, 4 vertical): Compare if view $v_1$ is aligned with $v_2$, such that one of the 4 distance functions is 0, e.g. $d_{ll}(v_1, v_2) = 0$

- For centering (2 features: 1 horizontal, 1 vertical): Compare if view $v_1$ is centered in $v_2$, such that $d_{ll}(v_1, v_2) = -d_{rr}(v_1, v_2)$.

- For overlaps (4 features: 2 horizontal, 2 vertical): Check if the $v_1$ and $v_2$ can possibly intersect, i.e. have overlapping x-coordinates in the horizontal case: $v_1.x_l \geq v_2.x_l$ and $v_1.x_l \leq v_2.x_r$.

- For the same size (2 features): Compare if the height or width difference of $v_1$ and $v_2$ is equal to 0, such that $s_w(v_1, v_2) = 0$ and $s_h(v_1, v_2) = 0$.

- For the same ratio (1 feature): Compare if the ratio $v_1$ and $v_2$'s aspect ratios is 1, such that $r(v_1, v_2) = 1$.

Table 5: Accuracy of different models (used as neural oracle) evaluated on the test dataset of $\mathcal{D}_{S+}$. For the pairwise accuracy we count as correct if the score of the positive sample is smaller than the score of the negative sample. In square brackets, we report the model performance if we additionally consider equal scores of the positive and the negative sample as correct (in case it differs).

| Training Dataset | Model | Pairwise Accuracy | | | |
|---|---|---|---|---|---|
| | | MLP | CNN | RNN | RNN + CNN |
| $\mathcal{D}_S$ | $f(\checkmark \mid x^*, y^*)$ | 84.4 [91.6]% | 80.0% | 78.5 [80.0]% | 83.5% |
| $\mathcal{D}_{S+}$ | $f(\checkmark \mid x^*, y^*, \mathcal{I})$ | 90.5 [97.6]% | 84.3% | 95.6% | 96.8% |
| | | Accuracy One vs. Many | | | |
| $\mathcal{D}_S$ | $f(\checkmark \mid x^*, y^*)$ | 44.5 [73.0]% | 36.7% | 36.3 [38.3]% | 55.8% |
| $\mathcal{D}_{S+}$ | $f(\checkmark \mid x^*, y^*, \mathcal{I})$ | 54.6 [87.3]% | 51.2% | 80.6 [81.0]% | 85.9% |

## B  EXPERIMENTS

In this appendix, we evaluate the oracle performance and include an ablation study of our models.

### B.1  ORACLE EVALUATION

We evaluate the oracle's performance (shown in Table 5) using two metrics: The pairwise accuracy and one vs. many accuracy. The pairwise accuracy is the binary ranking problem – for each pair consisting of a positive and negative candidate, it is considered to be correct, if the score of the positive example is higher than of the negative example (the pair was ranked correctly). To capture if the network can't distinguish between the samples, we also report a second value in square brackets in which we count as correct if the score of the positive example is higher or equal to the score of the negative example. In addition, we measure the one vs. many accuracy, in which the oracle is correct if the score of the correct candidate was higher than the scores of *all* incorrect candidates (since selecting the correct candidate out of many candidates represents what we are interested in during the end-to-end synthesis experiment).

Usually, models with a higher pairwise accuracy also have a higher one vs. many accuracy since more pairs are ranked correctly. The one vs. many accuracy might be lower than expected if the mistakes of the pairwise accuracy spread over a wide number of different samples (instead of having many mistakes in a small set of samples).

In general, the results of the one vs. many accuracy correlate with the results of the end-to-end synthesis. For example, RNN+CNN and RNN trained on $\mathcal{D}_{S+}$ achieve the highest scores in both of the experiments. One exception is the MLP model trained on $\mathcal{D}_{S+}$ dataset which performs worse on the end-to-end synthesis experiment even though the oracle performs better. A reason for this is that the MLP often makes mistakes when predicting the early views which then affect the positioning of the subsequent views. For the same reason, the accuracies of the oracle experiment are in general higher than the accuracy we observe in the synthesis experiment. Another interesting insight is that the MLP often scores the positive and the negative examples the same since there is a large difference in the pairwise accuracy depending on if we count the same score as correct or not. This shows that the MLP's feature functions are not expressive enough.

### B.2  ABLATION STUDY

To investigate which high-level properties are learned by our models, we generate an ablation dataset from the test dataset of $\mathcal{D}_{S+}$. The results of the study are collected in Table 6 and Table 7.

Recall from Section 4 that $\varphi_{\text{MLP}}(x, y)$ computes a vector of size 30 with the handcrafted functions which are described in detail in Appendix A.1. For each pair of positive $(x^*, y_{pos})$ and negative examples $(x^*, y_{neg})$ and their corresponding sample in the input specification $(x, y)$ we compute the difference of their handcrafted feature functions $d = [\varphi_{\text{MLP}}(x^*, y_{pos}) - \varphi_{\text{MLP}}(x, y)]_{(x,y)\in\mathcal{I}} -$

Table 6: Ablation study based on different subsets of the $\mathcal{D}_{S+}$ test dataset to check which general properties the model has learned. The $f(\checkmark \mid x^*, y^*, \mathcal{I})$ models are trained on $\mathcal{D}_{S+}$.

| | Pairwise Accuracy | | | | |
| Dataset | MLP | CNN | RNN | RNN + CNN | samples |
| --- | --- | --- | --- | --- | --- |
| Test | 90.5 [97.6] | 84.3 | 95.6 | 96.8 | all |
| Aspect ratio 1.0 | 96.9 | 63.5 | 92.7 | 99.0 | 96 |
| Horiz. centering-view | 100.0 | 89.0 | 96.6 | 98.2 | 5262 |
| Horiz. centering-views | 99.7 | 89.0 | 96.5 | 98.0 | 6360 |
| Vert. centering-view | 98.5 | 83.1 | 97.4 | 96.8 | 852 |
| Vert. centering-views | 98.2 | 89.3 | 97.1 | 97.7 | 6710 |
| Inside screen | 100.0 | 98.0 | 98.8 | 99.0 | 3292 |
| Intersections | 99.5 | 93.3 | 97.3 | 98.2 | 2792 |
| Horizontal alignment | 99.6 | 85.5 | 95.7 | 96.9 | 1658 |
| Vertical alignment | 99.7 | 95.0 | 98.0 | 98.6 | 3804 |
| Horizontal margins | 98.5 | 91.1 | 97.2 | 98.2 | 5642 |
| Vertical margins | 97.1 | 88.0 | 97.6 | 98.0 | 4547 |
| Specific aspect ratio | 100.0 | 61.1 | 100.0 | 100.0 | 18 |
| Same dimensions | 100.0 | 85.8 | 97.0 | 96.6 | 471 |

$[\varphi_{\texttt{MLP}}(x^*, y_{neg}) - \varphi_{\texttt{MLP}}(x, y)]_{(x,y)\in\mathcal{I}}$. We add a pair to the ablation dataset for the violated property if $d \neq 0$, that is, if the property on the positive sample is different from the negative sample. For example, if the number of view intersections of the positive sample and input specification are the same, but different for the negative sample, we add this pair of positive and negative samples to the intersections-ablation dataset.

The MLP models perform very well on all the ablation properties. The reason is that MLP's features are the handcrafted features which are designed to capture the properties in the ablation dataset. The MLP's performance on the whole test dataset is worse, which indicates that some properties in the dataset are not captured by the handcrafted features.

The CNN fails at preserving the aspect ratios due to the pooling and the downsampling operations. Alignments and intersection properties can be learned with local filters detecting overlapping views. Off-screen views can be detected by checking the downsampled representation after the convolutional layers. On these three properties, the CNN performs best. The RNN performs better on the ablation dataset than the CNN since it processes the exact numeric values of the view coordinates.

The overall performance on the ablation dataset is better than the one on the whole test dataset for all models, indicating that there are some features not expressed in the handcrafted functions. Furthermore, the results of the ablation study should be considered carefully: the ablation study shows tendencies, but features could be correlated or have common hidden features.

Table 7: Ablation study based on different subsets of the $\mathcal{D}_{S+}$ test dataset to check which general properties the model has learned. The $f(\checkmark \mid x^*, y^*)$ models are trained on $\mathcal{D}_S$.

| Dataset | Pairwise Accuracy | | | | |
| --- | --- | --- | --- | --- | --- |
| | MLP | CNN | RNN | RNN + CNN | samples |
| Test | 84.4 [91.6] | 80.0 | 78.5 [80.0] | 83.5 | all |
| Aspect ratio 1.0 | 85.4 | 54.2 | 65.6 | 70.8 | 96 |
| Horiz. centering-view | 97.5 | 86.7 | 88.0 | 89.5 | 5262 |
| Horiz. centering-views | 96.3 | 86.6 | 87.8 | 89.4 | 6360 |
| Vert. centering-view | 91.4 | 76.0 | 85.9 | 78.6 | 852 |
| Vert. centering-views | 93.0 | 86.6 | 84.9 | 89.6 | 6710 |
| Inside screen | 100.0 | 97.7 | 96.2 | 99.5 | 3292 |
| Intersections | 97.0 | 90.6 | 93.3 | 96.7 | 2792 |
| Horizontal alignment | 91.0 | 83.1 | 87.7 | 88.1 | 1658 |
| Vertical alignment | 96.4 | 90.7 | 85.6 | 97.3 | 3804 |
| Horizontal margins | 92.6 | 89.3 | 89.1 | 91.1 | 5642 |
| Vertical margins | 91.4 | 83.1 | 83.0 | 88.0 | 4547 |
| Specific aspect ratio | 100.0 | 100.0 | 61.1 | 44.4 | 18 |
| Same dimensions | 95.3 | 86.4 | 94.9 | 90.2 | 471 |

## C  VISUALIZATION

In this section, we provide visualizations of two concepts introduced in Section 4 – the rendered output candidates and how the CNN input is shifted for regularization. Further, we visualize the steps of the end-to-end synthesis and render outputs of synthesized programs.

### C.1  POSITIVE AND NEGATIVE OUTPUT CANDIDATES

Figure 4 shows four different rendered output candidates. The correct candidate is on the left and the three incorrect ones are on the right. The candidates differ in their 6th view which is moved around and overlaps in all of the three incorrect candidates. The three negative examples are generated by the synthesizer as described in Section 4 (for dataset $\mathcal{D}_{S+}$) or applying perturbations sampled from the synthesizer mistakes (for dataset $\mathcal{D}_S$).

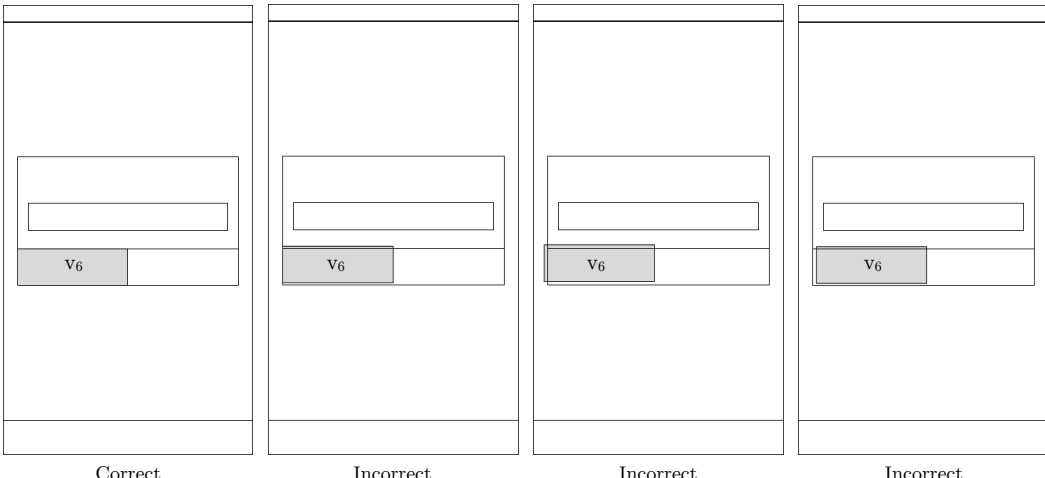

Figure 4: One positive example (left) and 3 negative examples (right).

## C.2 CNN REGULARIZATION

Figure 5 visualizes how the robustness of the CNN is increased by placing the input randomly within the input image as described in Section 4.

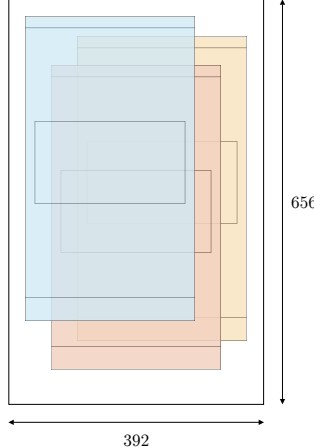

Figure 5: Illustration of three possible shifts (denoted by different colors) of the CNN input used during training.

## C.3 ADVANTAGE OF HAVING THE ADDITIONAL INPUT SPECIFICATION IN THE $\mathcal{D}_{S+}$ DATASET

Figure 6 visualizes the absolute view positions in the input specification on the left and the four candidates (differing in the red view) on the distinguishing device $x^*$ on the right. Looking at the four candidates, the first candidate is incorrect, since the red view is not centered. The third candidate is not the correct one either, since the red view intersects with the view below. It is hard to decide between the second and fourth candidate since the red view is centered in both cases. When taking the input specification into consideration, the second candidate is the correct one, since there is no space between the red view and the view below in candidate 4 (unlike in second candidate and the input specification).

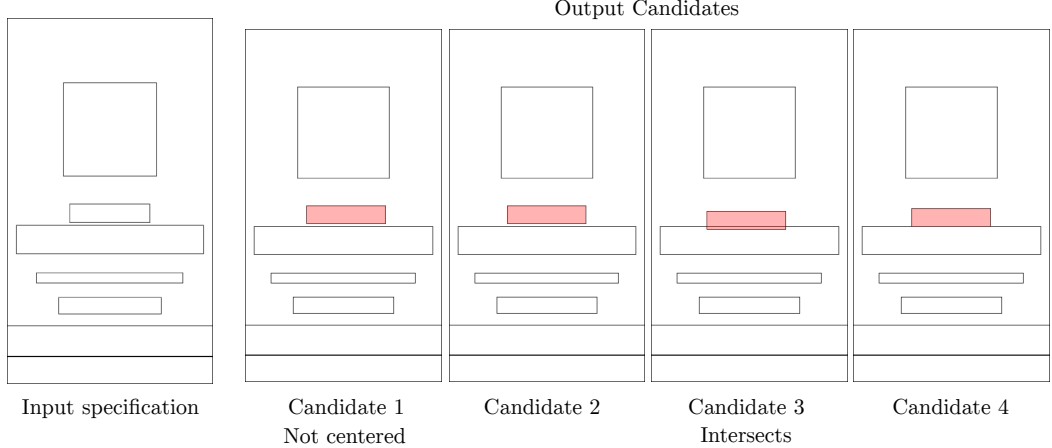

Figure 6: The input specification provides an important indication that candidate 2 is the correct one (unlike candidate 4).

### C.4 End-to-end synthesis procedure

In Figure 7 we present an example of the end-to-end synthesis described in Section 3 and Section 4. The input of the synthesizer is on the top left of the Figure 7 and consists of an input specification $\mathcal{I}$ with a single input-output pair $(x, y)$. The synthesis loop starts by the synthesizer generating a candidate program $p_1 \models \mathcal{I}$ (not shown) and checking whether ambiguities exists. To do this, it finds a *distinguishing input* $x^*$, in our example a smaller device $x^* = [0, 0, 1400, 2520]$ and a set of programs $p_2, \ldots, p_n$ that satisfy the input specification $\mathcal{I}$ but produce different outputs when evaluated on $x^*$. Three possible candidate outputs $p_1(x^*), p_2(x^*), p_3(x^*)$ are visualized in Figure 7 (column Iteration 1). Note that as described in Section 4, the synthesis proceeds by synthesizing the views iteratively therefore, the first iteration contains only the first view $v_1$. The oracle (in this example the CNN+RNN model trained on $\mathcal{D}_{S+}$) selects the second candidate (the oracle prediction is indicated by the dashed rectangle) since its score is the largest with 0.88. The position of $v_1$ on the device $x^*$ is added to the specification and the iterative synthesis proceeds by selecting the correct output for the second view. In the second iteration we perform the same steps, except now we synthesize different candidate outputs for the second view $v_2$ on the device $x^*$, given that the position of the first view $v_1$ is fixed from the previous iteration. The same process is repeated until the absolute positions of all the views are predicted for the distinguishing input $x^*$. The new input specification, shown at the bottom in Figure 7, contains the absolute view positions for both the original input $x$ as well as the distinguishing input $x^*$. After that, the whole process repeats by querying for another distinguishing device and extending the input specification even further.

Note that in our evaluation we restrict the distinguishing input to be among those device sizes for which we trained a model. Since our datasets consist of three different screen sizes the choice is small and all the experiments in Section 5 are evaluated with a single distinguishing input $x^*$. However, we could generate more than three dimensions for the training dataset and use them to train additional models. This was however not needed as three different devices are typically enough to resolve majority of ambiguities (as long as the oracle does not make mistakes).

#### C.4.1 Limitations of the MLP features

Figure 8 visualizes eight candidate outputs in the first synthesis iteration for the same application as in Figure 7. The MLP oracle assigns a score of 0 to all the output candidates in which view $v_1$ is off-screen. However, the handcrafted features are not expressive enough to distinguish between the candidate outputs in the bottom row which all have a score of 0.76. In general, the MLP performs bad in selecting the first views (earlier iterations) since there are only few views from which properties e.g. margins can be extracted. On the bottom right, the synthesis result is compared to the correct output. Since the first view was not selected correctly in the synthesis, all the subsequent inner views are also slightly off in comparison to the correct output.

### C.5 Examples of Synthesis Outputs

Figure 9 visualizes the rendered output of the synthesizer for the distinguishing device $x^*$. The correct output is on the left while the synthesis output of the different oracles is on the right. At first glance, the outputs of the MLP look visually appealing since the handcrafted features contain properties like intersections or off-screen and the model learns to avoid them. However, it is often not expressive enough, since the view sizes or positions are often different to the correct output. This happens in particular if there are only few views which is why the second view is often misplaced which leads to consecutive errors. This also explains the bad performance in the synthesis experiment in Table 1 in comparison to the oracle experiment in Table 5. The CNN outputs look much worse and often contain views which are misaligned or not centered. The RNN performs quite well one the first and third application but fails on the second application. The RNN+CNN predicts the correct output for the first application but fails on five views (out of 22) in the second and on seven (out of 17) views in the third application.

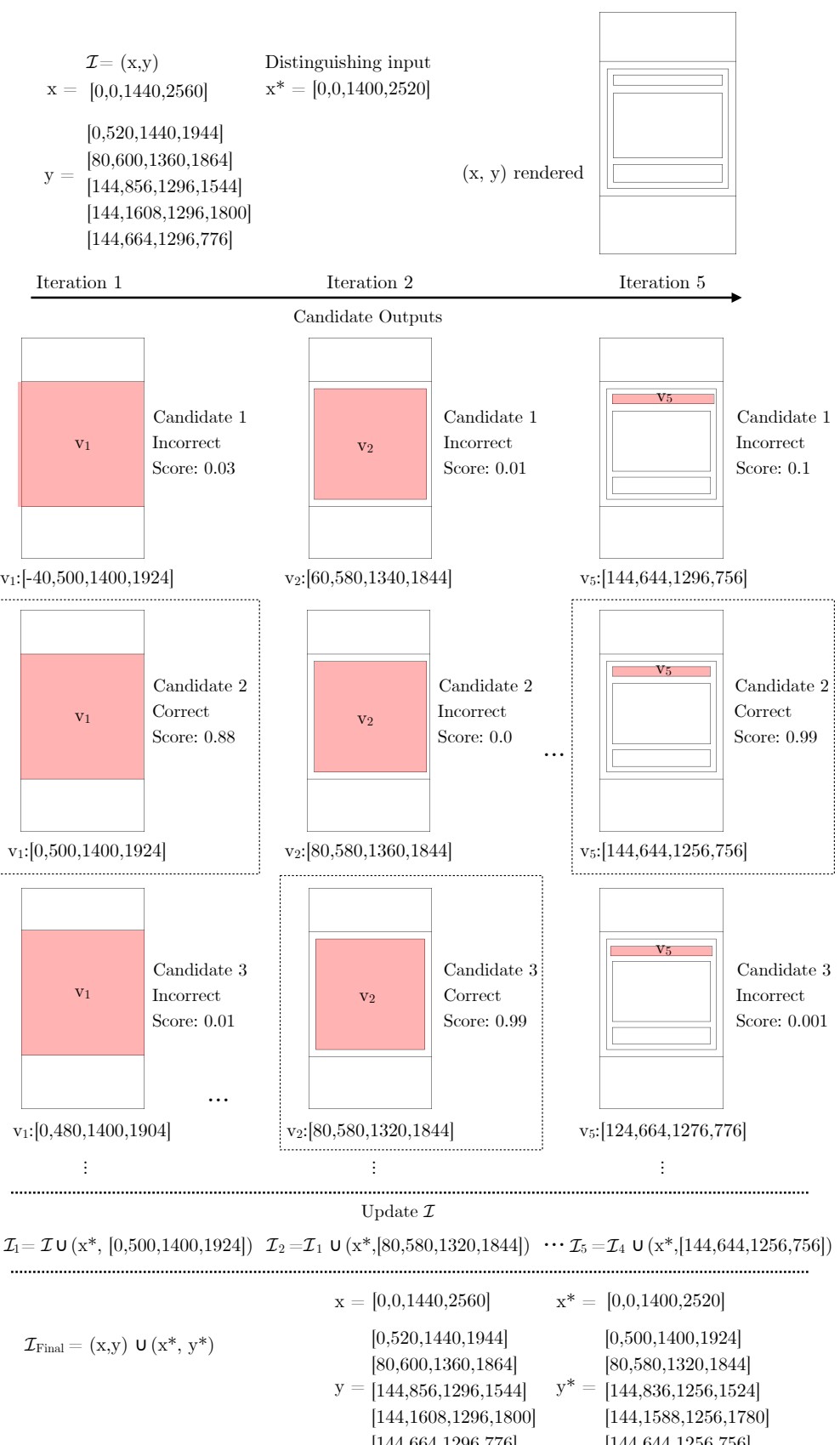

Figure 7: The synthesis procedure step-by-step.

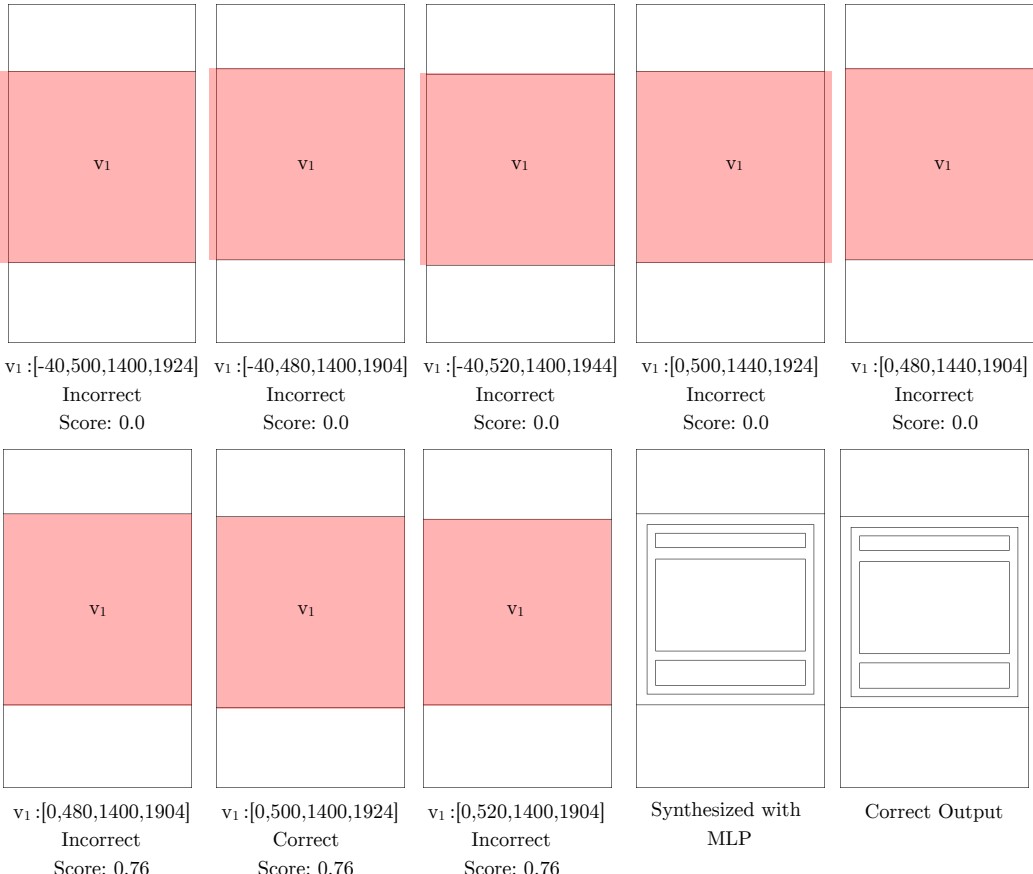

Figure 8: Eight output candidates in the first synthesis iteration. The oracle for the synthesizer was an MLP trained on $\mathcal{D}_{S+}$.

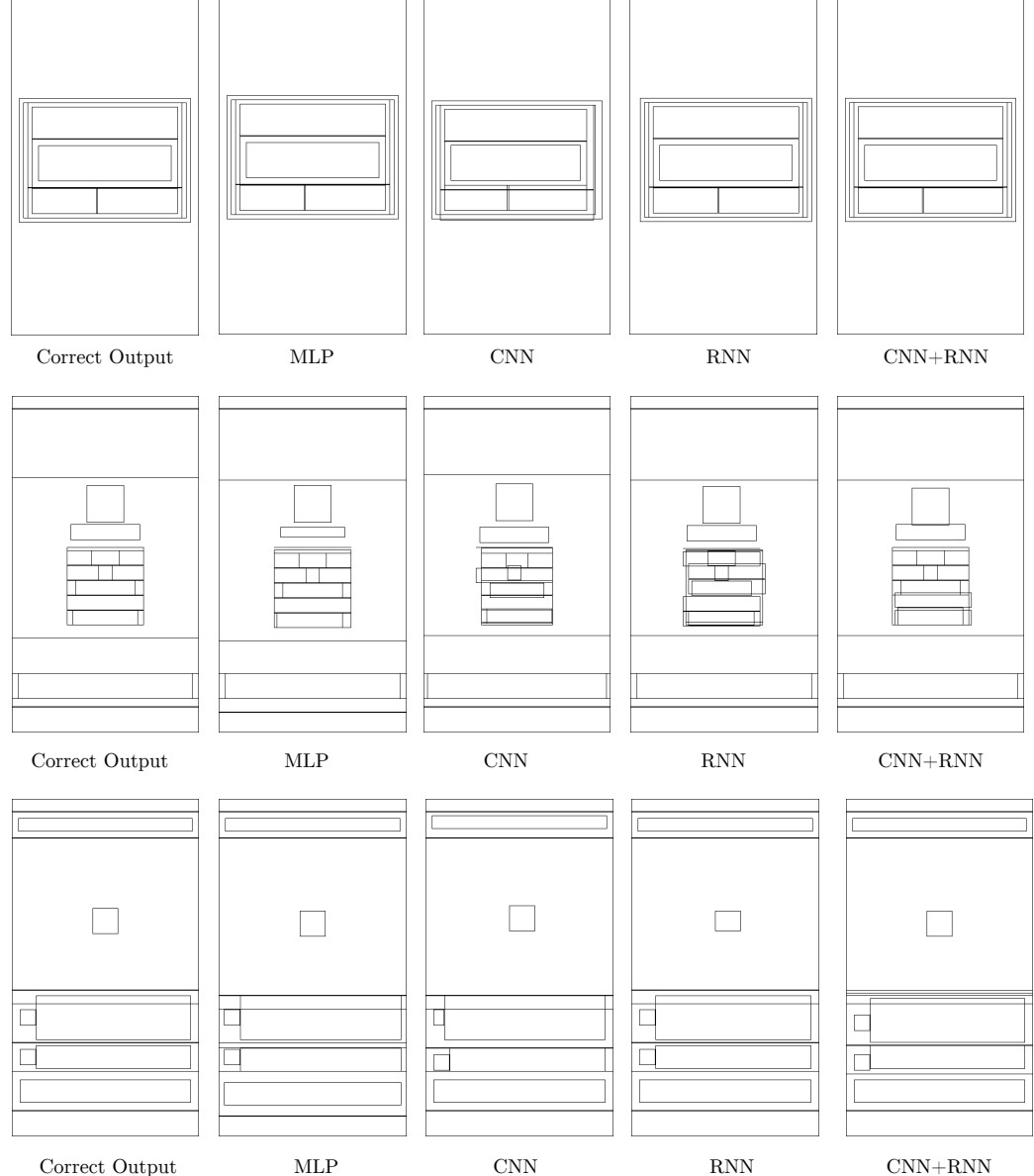

Figure 9: Rendered outputs on device $x^*$ of the synthesizer for different oracles which were trained on $\mathcal{D}_{S+}$.

