# OpenReview forum: "Guiding Program Synthesis by Learning to Generate Examples"
_ICLR.cc/2020/Conference — Accept (Poster)_

### Official Review · AnonReviewer1 · 2019-10-22
**Official Blind Review #1**

**Rating:** 8

**Review:**

= Summary
A method for a refinement loop for program synthesizers operating on input/ouput specifications is presented. The core idea is to generate several candidate solutions, execute them on several inputs, and then use a learned component to judge which of the resulting input/output pairs are most likely to be correct. This avoids having to judge the correctness of the generated programs and instead focuses on the easier task of judging the correctness of outputs. An implementation of the idea in a tool for synthesizing programs generating UIs is evaluated, showing impressive improvements over the baseline.

= Strong/Weak Points
+ The idea is surprisingly simple and applies to an important problem in program synthesis.
+ The experiments show that the method works very well in UI-generation domain
- The paper repeatedly claims general applicability to program synthesizers, but is only evaluated in the specific domain of UI-generating programs. I have substantial doubts that the approach would work as well in the domains of, e.g., string manipulation, Karel, or data structure transformations. My doubts are based on the fact that there are easily generalizable rules for UIs (no overlaps, symmetry, ...), whereas other domains are less easily described. This creates a substantial gap between paper claims and empirical results.
- The writting is somewhat sloppy (see below), which makes it sometimes hard to understand. Names such as "views" are used without explanation, and it's not explained how a device is an input to a program (yes, I get what this means, but it makes in unnecessarily hard to follow the paper)

= Recommendation
I would ask the authors to rewrite their paper to make less general claims, but believe that the general idea of judging the correctness of a program (or policy) by evaluating it on different inputs is a powerful concept that would be of substantial value to the wider ICLR audience. Improving the readability of the paper would make me improve my rating to a full accept.

= Minor Comments
* page 1, par "Generalization challenge": The second sentence here is 4 lines long and very hard to follow. Please rephrase.
* page 2, par 2: "no large real-word datasets exists" -> exist
* page 2, par 3: "even when both optimizations of InferUI are disabled": at this point, the reader doesn't know about any optimizations of InferUI.
* page 4, par 1: "i.e., $\exists p \in \mathcal{L}$" - $\mathcal{L}$ is undefined here (will be defined later on the page)
* page 4, par 2: "Generate a candidate program $p_1 \models \mathcal{I}$" - in step 2, there are suddenly also $p_2 \ldots p_n$, which are never explicitly generated. Either adapt this step, or explicitly generate them in step 2 based on the distinguishing input
* page 7 par 2: "We use one screen dimension as the input specification $\mathcal{I}$, the second as the distinguishing input" - this confused me, as the paper discussed discovering the distinguishing input (page 4, paragraph "Finding a distinguishing input"), whereas it sounds here like that input is manually selected.

**Experience Assessment:**

I have published in this field for several years.

**Review Assessment: Checking Correctness Of Derivations And Theory:**

N/A

**Review Assessment: Checking Correctness Of Experiments:**

I assessed the sensibility of the experiments.

**Review Assessment: Thoroughness In Paper Reading:**

I read the paper thoroughly.

---

> ### Author Response · Authors · 2019-11-09
> **Response to Reviewer #1**
>
> We thank the reviewer for the comments and suggestions to improve our paper.
>
> We are currently preparing a revision of our work that addresses both main suggestions: (i) to make less general claims and explicitly say the focus is on the domain of Android layouts, and (ii) to improve the readability of the paper. We will publish a list of changes, after also incorporating the feedback received from the other reviewers, once the revision is ready.
>
> Q: page 7 par 2: "We use one screen dimension as the input specification , the second as the distinguishing input" - this confused me, as the paper discussed discovering the distinguishing input (page 4, paragraph "Finding a distinguishing input"), whereas it sounds here like that input is manually selected.
>
> A: In our evaluation we restrict the distinguishing input to be among those device sizes for which we trained a model. Since our datasets consist of three different screen sizes the choice is small. However, we could generate more than three dimensions for the training dataset and use them to train additional models. This was however not needed as three different devices are typically enough to resolve majority of ambiguities (as long as the oracle does not make mistakes).

---

> > ### Comment · AnonReviewer1 · 2019-11-13
> > **Discussion of Review#1**
> >
> > Thank you for your clarifications in the paper, these satisfy me to update my score. Regarding my question on the search for a distinguishing input, I think it would be helpful to expand the description in Sect. 5 to explicitly state that the search for a distinguishing input is _not_ required in this setting.
> >
> > Minor comments in revision:
> > * page 1 par 2: "only a small number of them generalizes" -> generalize
> > * page 4 par 1: "user provided" -> "user-provided"

---

### Official Review · AnonReviewer3 · 2019-10-24
**Official Blind Review #3**

**Rating:** 3

**Review:**

[Summary]
This paper aims to improve the generalization of program synthesis, ensuring that the synthesized programs not only work on observed input/output (I/O) examples but also generalize well to assessment examples (i.e. model the real intent of the end-user). To this end, the paper proposes a framework that iteratively alternates between producing programs with an existing program synthesizer and augmenting examples to disambiguate possible programs with a neural oracle that learns to select correct outputs. Several architectures and the design of input of the neural oracle have been investigated. The experiments on Andriod layout program synthesis with an InferUI synthesizer show that the proposed framework can improve the generalization of synthesized programs. However, I find it is difficult to evaluate the effectiveness without sufficient qualitative results and the intermediate outputs (e.g. a distinguishing input and candidate outputs) of the proposed framework (see details below).

Significance: are the results significant? 4/5
Novelty: are the problems or approaches novel? 4/5
Evaluation: are claims well-supported by theoretical analysis or experimental results? 3/5
Clarity: is the paper well-organized and clearly written? 4/5

[Strengths]

*motivation*
- The motivation for improving the generalization of program synthesis by augmenting examples is convincing.

*novelty*
- The idea of utilizing a neural network to select correct outputs to augment examples for disambiguating the possible programs is intuitive and convincing. This paper presents an effective way to implement this idea.

*technical contribution*
- The paper investigates a set of network architectures and ways to specify the network input for learning the neural oracle. The RNN+CNN model that leverages both rendered views and features seems effective.

*clarity*
- The overall writing is clear. The authors utilize figures well to illustrate the ideas. Figure 1 clearly shows the proposed framework.

*experimental results*
- The presentations of the results are clear. The results demonstrate that the proposed framework can improve generalization accuracy.

*reproducibility*
- Given the clear description in the main paper and the details provided in the appendix, I believe reproducing the results is possible if the dataset is available.

[Weaknesses]

*related work*
The descriptions of the related work are not comprehensive. Some neural program synthesis works explore a variety of mechanisms to encode examples and fuse their features, which are not mentioned in the paper. [Devlin et al. in ICML 2017] investigates different attention mechanisms to sequentially encode a set of I/O examples and performs pooling to merge them. [Sun et al. in ICML 2018] proposes a doubly encoding method to capture more details of examples and merge the features using a relation network. I believe it would be interesting to see if these methods could further improve the performance of the neural oracle.

*experiment setup*
- The experiments are not sufficient. While the claims look promising, the proposed method is only evaluated in only one dataset, which is not sufficiently convincing. I suggest the authors to also experiment the FlashFillTest dataset where string transformation programs are synthesized.
- A more comprehensive description of the dataset is lacking.

*experiment results*
- I find it hard to judge the effectiveness of the proposed framework without seeing sufficient qualitative results. I suggest the authors randomly sample some synthesized programs (both success and failure) and present them in the paper.
- I believe it is important to present some examples of the given I/O pairs, initially synthesized programs (p_1), found distinguishing input (x*), candidate outputs (y), the prediction of the neural oracle (i.e. selected outputs), the augmented examples (I \cup {(x*, y*)}), and finally the next synthesized program. Without this, it is very difficult to understand the performance of the proposed framework and what could go wrong.

*ablation study: the neural oracle*
Only the final performance (i.e. the program synthesis performance) is shown in the paper. I believe it would be helpful if the performance of the neural oracle was also presented. As the whole framework depends on how accurate the neural oracle can select the correct output, it is important to evaluate this. One way to show this is to simply show the performance of all the neural oracles (with different architectures) trained on D_S (the positive samples and the incorrect samples) or even D_{S+}.

Devlin et al. "RobustFill: Neural Program Learning under Noisy I/O" in ICML 2017
Sun et al. "Neural Program Synthesis from Diverse Demonstration Videos" in ICML 2018

**Experience Assessment:**

I have published one or two papers in this area.

**Review Assessment: Checking Correctness Of Derivations And Theory:**

N/A

**Review Assessment: Checking Correctness Of Experiments:**

I carefully checked the experiments.

**Review Assessment: Thoroughness In Paper Reading:**

I read the paper thoroughly.

---

> ### Author Response · Authors · 2019-11-09
> **Response to Reviewer #3**
>
> We thank the reviewer for the thorough comments. Please find the answers to your questions below:
>
> *reproducibility*
> Given the clear description in the main paper and the details provided in the appendix, I believe reproducing the results is possible if the dataset is available.
>
> A: We are happy to hear that. We are in the process of releasing the code, datasets and trained models.
>
> Q: Could you provide a more comprehensive description of the dataset?
>
> A: The datasets $\mathcal{D}_{U}$ and $\mathcal{D}_{S}$ are obtained by using the screenshot and UI metadata collected as part of the RICO dataset. A thorough description of the dataset including the examples is available at http://interactionmining.org/rico. For the $\mathcal{D}_{S+}$ we downloaded the same set of applications used in the RICO dataset and rendered them on three different devices. We will include all the datasets and their description as part of our code release.
>
> *experiment setup*
> Q: The experiments are not sufficient. While the claims look promising, the proposed method is only evaluated in only one dataset, which is not sufficiently convincing.
>
> A: We believe focusing on a single real-world type of data which is important in practice (UIs) is already a challenging task, and similarly to other real-world synthesizers (e.g., FlashFill), we did not experiment with a different domains. As mentioned, we are happy to update the write-up to explicitly say the focus is on one type of domain. We do note that considering a real-world dataset is already in contrast to most other existing datasets used in program synthesis that are either very small/toy or contain only synthetic examples.
>
> Q: I suggest the authors to also experiment the FlashFillTest dataset where string transformation programs are synthesized.
>
> A: Unfortunately, to our best knowledge this dataset is not publicly available as discussed here (https://openreview.net/forum?id=Skp1ESxRZ).
>
> *experiment results*
> Q: Would it be possible to randomly sample some synthesized programs (both success and failure) and present them in the paper.
>
> A: Yes, we are working on a revision of our paper that will include such examples.
>
> Q: Would it be possible to present some examples of the given I/O pairs, initially synthesized programs ($p_1$), found distinguishing input ($x^*$), candidate outputs ($y$), the prediction of the neural oracle (i.e. selected outputs), the augmented examples ($\mathcal{I} \cup {(x^*, y^*)}$), and finally the next synthesized program.
>
> A: Yes, we are working on a revision of our paper that will include such examples.
>
> *ablation study*
> Q: Only the final performance (i.e. the program synthesis performance) is shown in the paper. I believe it would be helpful if the performance of the neural oracle was also presented. One way to show this is to simply show the performance of all the neural oracles (with different architectures) trained on $\mathcal{D}_{S}$  (the positive samples and the incorrect samples) or even $\mathcal{D}_{S+}$ .
>
> A: We are happy to include these results in the appendix. Below we provide the summary of the neural oracle results when evaluated on the $\mathcal{D}_{S+}$ dataset:
>
> Trained on $\mathcal{D}_{S}$: MLP: 44.5%, RNN: 51.2%, CNN: 36.6%, RNN+CNN: 58.7%
> Trained on $\mathcal{D}_{S+}$: MLP: 54.6%, RNN: 87.5%, CNN 51.2%, RNN+CNN: 89.9%.
>
> In general, the neural-oracle results correlate with the results of the end-to-end synthesis. One exception is the MLP model trained on $\mathcal{D}_{S+}$  dataset which performs worse on the end-to-end synthesis experiment even though the oracle performs better. A reason for this is that the MLP often makes mistakes when predicting the early views which then affects predictions for subsequent views. For the same reason, accuracies of the oracle experiment are in general higher than the accuracy we observe in the synthesis experiment.
>
> We will provide these additional results and the discussion in the updated version of our paper.
>
> *related work*
> Q: Some neural program synthesis works explore a variety of mechanisms to encode examples and fuse their features, which are not mentioned in the paper [Devlin et al. in ICML 2017, Sun et al. in ICML 2018]. I believe it would be interesting to see if these methods could further improve the performance of the neural oracle.
>
> A: We already included a comparison to the work of Devlin et.al and are happy to also include comparison to the work of Sun et. al. In our related work we tried to focus on the high level approaches used by prior works to resolve ambiguities in the input specification rather than the details of the neural architectures they use. Having said that, we do believe that it is possible to improve the performance of the neural oracle even further by using a more complex architecture, better datasets or via including additional information (e.g., encoding both the output and the synthesized programs).

---

> > ### Comment · AnonReviewer3 · 2019-11-13
> > **Re: Response to Reviewer #3**
> >
> > I appreciate the effort the authors put into revising the paper. The revision addresses my comments on  (1) evaluating the performance of different neural oracle models, (2) presenting intermediate results, and (3) missing related works.
> >
> >  I have a mixed feeling about accepting this paper. I like the intuitive idea of iteratively alternating between producing programs with an existing program synthesizer and augmenting examples to disambiguate possible programs with a neural oracle that learns to select correct outputs. However, I am still not completely satisfied with the fact that this paper only conducts experiments on a single dataset. I would like to see if the proposed framework can work on different program synthesis tasks. I decided not to update my score at this moment.

---

> > > ### Author Response · Authors · 2019-11-15
> > > **Evaluating on different tasks**
> > >
> > > We agree that it would be interesting to evaluate our approach on a different program synthesis task. However, this requires a non-trivial effort that often includes replicating the original work, obtaining suitable dataset for training and testing as well as designing a suitable neural architecture for the given domain at hand. For example, to obtain the Rico dataset (D_U dataset used in our work), Deka et.al. had to develop a crowdsourcing platform where workers spent 2, 450 hours over five months and got paid ~$20, 000 in compensation. Similarly, in our work it took over a month just to obtain the D_S+ dataset (which extends the original Rico dataset).
> > >
> > > Note that these challenges are not specific to our work as the majority of prior works also focus on a single task. We include a representative list of works below:
> > >
> > > Menon et. al. 2013, “A Machine Learning Framework for Programming by Example”,
> > >      - text processing task: handcrafted samples + obtained from Microsoft Excel help forums
> > >
> > > Devlin et. al., 2017  “RobustFill: Neural Program Learning under Noisy I/O”,
> > >     - string transformations (FlashFillTest dataset)
> > >
> > > Parisotto et. al. 2017, “Neuro-Symbolic Program Synthesis”,
> > >     - string transformations (FlashFill dataset)
> > >
> > > Ellis et. al., 2018, “Learning to Infer Graphics Programs from Hand-Drawn Images”,
> > >     - graphics programs
> > >
> > > Liang et. al, 2010, “Learning Programs: A Hierarchical Bayesian Approach”,
> > >     - text editing domain, illustration on a simple arithmetic domain
> > >
> > > Balog et. al., 2017, “DeepCoder: Learning to Write Programs”,
> > >     - restricted DSL for functional list processing
> > >
> > > Shin et. al., 2018, “Improving Neural Program Synthesis with Inferred Execution Traces”,
> > >     - Karel domain
> > >
> > > Bunel et. al. 2018, “Leveraging Grammar and Reinforcement Learning for Neural Program Synthesis”,
> > >     - Karel domain
> > >
> > > Singh, 2016, “BlinkFill: Semi-supervised Programming By Example for Syntactic String Transformations”,
> > >     - string transformations (FlashFillTest dataset)
> > >
> > > Singh et. al., 2015, “Predicting a Correct Program in Programming By Example”:
> > >     - string transformations (FlashFillTest dataset)
> > >
> > > Polosukhin, Skidanov, 2018, “Neural program search: Solving programming tasks from description and examples.”,
> > >     - AlgoLisp
> > >
> > > We also include small number of works that do evaluate on multiple tasks:
> > >
> > > Ellis et. al. 2017, “Learning to Learn Programs from Examples: Going Beyond Program Structure”):
> > >      - string transformation,
> > >      - text extraction
> > >
> > > Nye et. al. 2019, “Learning to Infer Program Sketches”:
> > >      - list processing problems,
> > >      - string transformations,
> > >      - AlgoLisp

---

> > > > ### Comment · AnonReviewer3 · 2019-11-15
> > > > **Re: Evaluating on different tasks**
> > > >
> > > > I appreciate the authors for pointing out the difficulty of obtaining a suitable dataset for program synthesis. However, I am entirely not convinced by this list. First of all, while many papers here use a single dataset, most of the papers use a commonly acceptable dataset such as Karel (not really a fixed dataset but a domain), FlashFillTest, and AlgoLisp. Yet, this submission uses a different dataset that has not been explored before, which makes it harder to compare with other methods. On the other hand, I would like to note that not all the common practices shared by many publications are good, and certainly should not prevent the authors from further improving this submission. Therefore, I still consider this as a weakness but do appreciate the overall efforts.

---

### Official Review · AnonReviewer2 · 2019-10-24
**Official Blind Review #2**

**Rating:** 8

**Review:**

This paper handles the challenge of generating generalizable programs from input-output specifications when the size of the specification can be quite limited and therefore ambiguous. When proposed candidate programs lead to divergent outputs on a new input, the paper proposes to use a learned neural oracle that can evaluate which of the outputs are most likely. The paper applies their technique to the task of synthesizing Android UI layout code from labels of components and their positions.

The experiments compare the method against the baseline InferUI. To summarize the results, we can see that the proposed method in the paper can perform about as well as existing hand-crafted constraints that guide the search process of the previous work, when training an oracle on the dataset with negative examples created by noising the positive examples.

One limitation of the method is that it would works best when there is a clear latent structure behind the outputs produced by the correct program, such as in the paper's target domain of generating UIs where there are clear aesthetic rules and design guidelines that make it possible to evaluate which output is most preferred. For other domains, it may be more important to evaluate the candidate program together with its output, which would make it similar to a re-ranking approach.

I believe this paper presents a novel and insightful approach to creating programs from imprecise specifications. Therefore, I vote to accept the paper.

Some questions for the authors:
- How big was $\mathcal{I}_i$ in the supervised dataset $\mathcal{D}_{S+}$? Was it always 3?
- I wasn't able to find any evaluation of a model trained on $\mathcal{D}_U$, did I miss it in the paper?

**Experience Assessment:**

I have published one or two papers in this area.

**Review Assessment: Checking Correctness Of Derivations And Theory:**

I assessed the sensibility of the derivations and theory.

**Review Assessment: Checking Correctness Of Experiments:**

I assessed the sensibility of the experiments.

**Review Assessment: Thoroughness In Paper Reading:**

I read the paper at least twice and used my best judgement in assessing the paper.

---

> ### Author Response · Authors · 2019-11-09
> **Response to Reviewer #2**
>
> We thank the reviewer for the comments and clarifying questions.
>
> Q: One limitation of the method is that it works best when there is a clear latent structure behind the outputs produced by the correct program, such as in the paper's target domain of generating UIs. For other domains, it may be more important to evaluate the candidate program together with its output, which would make it similar to a re-ranking approach.
>
> A: We agree, there are domains (like the one considered in our work) where the output is much more important and has much more structure than the program used to generate it. Similarly, there are other domains where the opposite is true. We note that it would be interesting to try our approach even in the latter case where the additionally generated input-output example could be used to refine the model confidence (or as a sanity check, especially if complex neural models are used)
>
> Q: How big was $\mathcal{I}_i$ in the supervised dataset? Was it always 3?
>
> A: For all our synthesis experiments, the input specification always contained only a single example (containing a single device and its associated views). This is because this corresponds to how the tool would typically be used by a developer. The fact that the input specification in the dataset $\mathcal{D}_{S+}$ contains multiple samples (indeed, it contains 3 samples per application for the different devices) was used only during oracle training.
>
> Q: I wasn't able to find any evaluation of a model trained on $\mathcal{D}_{U}$, did I miss it in the paper?
>
> A: We omitted such evaluation in the paper as in our domain the dataset $\mathcal{D}_{S}$ is created synthetically from $\mathcal{D}_{U}$ (by automatically generating negative samples) and results in models that are strictly better than those trained only on $\mathcal{D}_{U}$.

---

> > ### Comment · AnonReviewer2 · 2019-11-14
> > **Response**
> >
> > Thank you for the response and the clarifications!

---

### Author Response · Authors · 2019-11-12
**Paper Revision**

Dear reviewers, we would like to thank you for all your comments and suggestions.
We have updated our paper with a revision to address them. We summarize the main changes below:

1) [Introduction] Make less general claims and explicitly say that the focus is on the domain of Android layouts:

We have rewritten the introduction section to remove claims of generalizability and to explicitly say that the domain considered in our work is Android layouts. We also remove the example discussing Excel spreadsheets.

2) [Related Work] The related work now includes that approaches such as Devlin et al. 2017, Sun et al. 2018 and Parisotto et.al 2017, which propose neural architectures that encode input-output examples, might lead to further improvements to the neural models designed in our work. We have also added references to the related line of work on neural machines (e.g, Graves et. al. 2016)

3) Incorporate various comments and suggestions throughout the paper to improve readability.

4) [Appendix B.1] Additional evaluations of all the learned neural oracle models

5) [Appendix B.2] Additional ablation study that investigates how well our models learn high level handcrafted properties derived from InferUI

6) [Appendix C.3] Example and a discussion illustrating the advantage of training on the dataset D_S+ compared to D_S

7) [Appendix C.4] Example of the individual steps performed during the synthesis

8) [Appendix C.5] Visualization of good and bad synthesis outputs produced by the models used in our work

---

### Decision · Program_Chairs · 2019-12-19

**Decision:**

Accept (Poster)

**Comment:**

The paper consider the problem of program induction from a small dataset of input-output pairs; the small amount of available data results a large set of valid candidate programs.
The authors propose to train an neural oracle by unsupervised learning on the given data, and synthesizing new pairs to augment the given data, therefore reducing the set of admissible programs.
This is reminiscent of data augmentation schemes, eg elastic transforms for image data.

The reviewers appreciate the simplicity and effectiveness of this approach, as demonstrated on an android UI dataset.
The authors successfully addressed most negative points raised by the reviewers in the rebuttal, except the lack of experimental validating on other datasets.

I recommend to accept this paper, based on reviews and my own reading.
I think the manuscript could be further improved by more explicitly discussing  (early in the paper) the intuition why the authors think this approach is sensible:
The additional information for more successfully infering the correct program has to come from somewhere; as no new information is eg given by a human oracle, it was injected by the choice of prior over neural oracles.
It is essential that the paper discuss this.